# A Machine Learning Framework for Automated Accident Detection Based on Multimodal Sensors in Cars

**DOI:** 10.3390/s22103634

**Published:** 2022-05-10

**Authors:** Hawzhin Hozhabr Pour, Frédéric Li, Lukas Wegmeth, Christian Trense, Rafał Doniec, Marcin Grzegorzek, Roland Wismüller

**Affiliations:** 1Research Group of Operating Systems and Distributed Systems, University of Siegen, Hölderlinstr. 3, 57076 Siegen, Germany; roland.wismueller@uni-siegen.de; 2Institute of Medical Informatics, University of Lübeck, Ratzeburger Allee 160, 23538 Lübeck, Germany; fr.li@uni-luebeck.de (F.L.); christian.trense@student.uni-luebeck.de (C.T.); marcin.grzegorzek@uni-luebeck.de (M.G.); 3Intelligent Systems Group (ISG), University of Siegen, Hölderlinstr. 3, 57076 Siegen, Germany; lukas.wegmeth@uni-siegen.de; 4Department of Biosensors and Biomedical Signal Processing, Faculty of Biomedical Engineering, Silesian University of Technology, Roosevelta 40, 41-800 Zabrze, Poland; rafal.doniec@polsl.pl; 5Department of Knowledge Engineering, University of Economics in Katowice, Bogucicka 3, 40-287 Katowice, Poland

**Keywords:** automated accident detection, feature extraction, feature learning, time series processing, deep neural networks

## Abstract

Identifying accident patterns is one of the most vital research foci of driving analysis. Environmental or safety applications and the growing area of fleet management all benefit from accident detection contributions by minimizing the risk vehicles and drivers are subject to, improving their service and reducing overhead costs. Some solutions have been proposed in the past literature for automated accident detection that are mainly based on traffic data or external sensors. However, traffic data can be difficult to access, while external sensors can end up being difficult to set up and unreliable, depending on how they are used. Additionally, the scarcity of accident detection data has limited the type of approaches used in the past, leaving in particular, machine learning (ML) relatively unexplored. Thus, in this paper, we propose a ML framework for automated car accident detection based on mutimodal in-car sensors. Our work is a unique and innovative study on detecting real-world driving accidents by applying state-of-the-art feature extraction methods using basic sensors in cars. In total, five different feature extraction approaches, including techniques based on feature engineering and feature learning with deep learning are evaluated on the strategic highway research program (SHRP2) naturalistic driving study (NDS) crash data set. The main observations of this study are as follows: (1) CNN features with a SVM classifier obtain very promising results, outperforming all other tested approaches. (2) Feature engineering and feature learning approaches were finding different best performing features. Therefore, our fusion experiment indicates that these two feature sets can be efficiently combined. (3) Unsupervised feature extraction remarkably achieves a notable performance score.

## 1. Introduction

Identifying accident patterns is one of the most vital research foci of driving analysis. According to the global status report on road safety conducted by the World Health Organization (WHO) [1], the number of traffic-related fatalities continues to rise steadily.

Despite considerable improvements in road safety by programs such as “Driver Assistance”, “Safety Awareness Services” and “Automatic Crash Notification (ACN)” systems, accident detection and prevention in driving studies are still of crucial significance. For this reason, accident detection studies have drawn the attention of insurance and fleet management companies [2,3]. Environmental, road safety, and commercial applications such as insurance and loan qualifications are just a few fields where the contribution of accident analysis could be significant [4].

An important contribution of the accident detection studies is the post-crash applications concerning immediate dispatch of the emergency and roadside assistance services [5,6,7,8]. However, these studies do not analyze accident patterns but rather imply factors such as airbag deployment to detect an accident. A recent application of accident detection, which studies accident patterns, is automated boxes installed in cars that minimize the service and overhead inspection cost of fleet operators by detecting car damages due to minuscule accidents, bumps, accelerations and braking manoeuvres [9].

In this regard, there are very few studies concerning accident detection using machine learning (ML) [4]. The reasons for this are multifaceted. To begin with, in study cases such as accident detection, providing positive labeled data is nearly impossible. Moreover, preparing a large real-world data base containing accident events is not only expensive and time consuming, but also hindered by competitive automobile industry and data privacy issues. To the best of our knowledge, real-world accident data provided by the second Strategic Highway Research Program (SHRP2) Naturalistic Driving Study (NDS) [10] is the only large-scale data base (including vehicles time series) that has been collected so far. Data scarcity in the field of accident detection is an obstacle to obtaining an automated accident detection system trained using ML, since the latter requires a large data set to train models that should be inclusive/unbiased and of good quality. There can also be times where one must wait for new data to be generated. Finally, the complex nature of accident events themselves present enormous challenges. Study cases like these require manually labeled training data, where the exact conditions characterizing the definition of an accident need to be predefined. According to Virginia Tech Transportation Institute (VTTI) [10], an accident is defined as “any contact that the subject vehicle has with an object, either moving or fixed, at any speed in which kinetic energy is measurably transferred or dissipated. This also includes non-premeditated departures of the road, as well as instances where the subject vehicle strikes another vehicle, roadside barrier, pedestrian, cyclist, animal, or object on or off the roadway”. Based on the above definition, minor crashes where the tire is struck with little or no risk element (e.g., clipping a curb during a tight turn) could be considered as accidents. Such cases could only be noticed by reporting the incident or during an inchmeal inspection. These cases often pose a major challenge for automated accident detection as well.

Accident detection is usually translated into a binary classification problem, where input data are used to train models representing accident and non-accident classes. ML-based accident detection studies can be categorized depending on the type of data used to train their model. Two large categories can be identified, one relying on traffic data, the other on external sensors such as smartphones, acoustic sensors or cameras [4]. However, the performance of these kinds of prediction and detection systems is greatly confined by the availability of monitoring sensors, funds, weather, traffic flow, etc., while external sensors can end up being difficult to set up and unreliable, depending on how they are used [11]. For the aforementioned reasons, internal car data have been investigated as the third possibility for ML-based accident detection, which does not suffer from the abovementioned disadvantages. As a general remark, the term *car* in this paper refers to a *passenger vehicle*. Each modern car (manufactured starting from the mid-1990s), contains a variety of sensors that provide reliable time-series data for the most basic driving attributes such as speed, steering wheel angle and gas pedal position. This data can be of great advantage for accident detection and is readily available by monitoring the in-car network. For instance, the on-board diagnostics (OBD) adapter is the most available modality to capture driving patterns in network (e.g., CAN-Bus) protocols inside cars. More information about classifying and finding out the semantics of the most important signals transmitted on the in-car network can be found in our previous study in [12].

ML can be applied on in-car network data by following a standard framework usually referred to as the *pattern recognition chain* (PRC) that consists of four steps as illustrated in Figure 1. Firstly, during data acquisition, the chosen sensors based on the nature of the classes and their availability are to be considered. In the second step, data pre-processing includes operations such as sensor calibration, unit conversion, normalization and segmentation to make the data suitable for further analysis. Next, feature extraction is about deriving the most relevant information to the classes from each segment of data, thus leading to an abstracted and informative representation of each data segment. In the last step, classifiers are trained to separate different classes in the feature space, which in our case study would be distinguishing between accident and non-accident driving events.

Experience has shown that each of these steps has a significant impact on accident detection performance [13]. However, data acquisition and pre-processing are mostly dependent on the data itself, while classification relies on well-known state-of-the-art approaches whose reliability was proven in past studies. For these reasons, feature extraction is the step where the margin of improvement is the largest and which has been the most investigated in past ML studies. Accordingly, to provide a ML framework for accident detection, a study on applicable feature extraction approaches is of great significance.

Past feature extraction research concentrates on two types of feature extraction techniques: feature engineering and feature learning. On the one hand, feature engineering is a traditional feature extraction approach relying on prior expert knowledge of the data to propose manually crafted features adapted to the problem to be solved. Feature engineering, for instance, includes the so-called *handcrafted* method that computes very simple statistical values as features on the input data and/or designed by experts to solve a specific problem [14,15,16]. However, feature engineering is not always an optimal way of extracting features when the expert knowledge is not available, and there is no guarantee that the chosen features would be optimal. On the other hand, feature learning is an automated feature extraction process using artificial neural networks (ANNs) that are ML models that were shown to be able to extract highly efficient features, provided decent pre-processing and segmentation steps on the data [17] have been carried out. Despite the popularity of ANNs in obtaining efficient features, finding the proper parameters and properly training the model can be complicated. Additionally, the features learned by ANNs are usually difficult to interpret, which can be a major obstacle to their use in many applications. Therefore, providing a comprehensive study on applying feature engineering and feature learning approaches and analyzing the optimal feature extraction approaches in spite of their respective drawbacks is of tremendous importance. To summarize, the contributions of our paper are as follows:This paper is the first study investigating ML-based accident detection on basic in-car network data. Our work is a unique and innovative study on detecting real driving accidents from the most accessible and affordable data sources inside cars.This paper presents a detailed ML framework based on the PRC introduced in Figure 1 to perform accident detection using basic in-car network data. In addition, it uses this framework to provide a comparison of state-of-the-art ML feature extraction techniques, applicable on in-car sensor data for accident detection based on SHRP2 NDS crash data set providing gas-pedal position, speed, steering angle and acceleration sensors. Using this framework, we obtain very promising results for automated accident detection based on a naturalistic data set.

This paper is organized in the following manner: In Section 2, we present the related work. Section 3 describes the proposed ML framework illustrated in Figure 1, with a brief explanation of materials and the chosen techniques. Section 4 is about the implementation details of all applied algorithms. Section 4 discusses the results. In Section 5, the interpretation of results is performed. We conclude the paper in Section 6 by describing our accomplishments, study limitations and future work.

## 2. Related Work

Due to the increased need for mobility, driving behavior analysis applications have become an important area of research. The result of driving behavior analysis has significant importance for the automotive and intelligent transportation industry, automobile insurance and the government organizations controlling infrastructure and public transportation. Numerous works address the importance of driving behavior analysis in relation to traffic, safety and ecological concerns [18,19,20,21], whereas many others concern the driver’s behavior analysis [4,22]. Due to the diversity of research goals, applications, study contributions and data modalities, there is no specific study baseline or research categorization in the domain of driving behavior analysis specifically for accident detection. Therefore, to review the recent state-of-the-art on accident detection, an overview of the relevant works in the field of driving behavior analysis is needed. It can be noted that time-series feature extraction is a topic that is quite well explored [23,24]. However, the conclusions of such feature extraction studies do not seem to transfer from one application field to another, and this topic has remained relatively unexplored for accident detection. Two main categories of studies are presented in this section. The first category summarizes the state-of-the-art in “driving behavior analysis” in general. The second category is specific to “accident detection” literature.

### 2.1. Driving Behavior Analysis

Various past studies have surveyed the field of driving behavior analysis. Zinebi et al. [22] performed a literature review splitting driver behavior analysis work into three main sub-applications: (i) accident prevention, (ii) driving styles assessment and (iii) driver intent prediction. It also identifies three categories of methods used to solve the problem: 1. index systems, 2. image processing and 3. statistical methods and ML. The principle of index system methods is to define *indices*, i.e., metrics that can objectively quantify high-level concepts linked to driver behavior such as risk of accident, degree of consciousness, etc. An example is an index such as TTC (time to collision) defined by Mori et al. [25] to measure the time the vehicle spends before hitting an object in its environment. They used this index to calculate the environmental risk score at a given time. Based on the correlation between the two vectors of driver’s attention and the environmental risk score, they observed that the degree of awareness is generally higher for expert drivers than for non-expert drivers.

Image processing is another popular method for driving behavior analysis due to how easily available vision sensors have become. Detecting driver’s drowsiness using computer vision techniques from an iPhone and the GPS to track the position of the car is an example of an image processing application for driving analysis [2]. The last category of methods used in the reviewed literature in Zinebi et al. [22] is statistical and uses ML methods. Bachoo et al. [26] utilized multiple linear regression to explore the role of personality traits such as anger, impulsivity, etc., on reported actions of risky driving behavior using a cross-sectional questionnaire. A random forest (RF) classifier was used by Jahangiri et al. [27] to classify driver behavior into two categories: violation and compliance at signalized intersections. They utilized the distance to the intersection, velocity, acceleration, time to the intersection, required deceleration parameter and velocity-based handcrafted (HC) features acquired from radar, video cameras and signal phase sniffers at intersections. In their work, they predict driving violations at the signalized intersection with accuracies of 97.9% and 93.6% for the SVM and RF models, respectively. Ohn et al. [28] adopted support vector machine (SVM) to classify the driver’s activities based on his/her hand positions using cameras placed in the car. In their study, they used hand motions in terms of activity classification and prediction from naturalistic driving images. Image-based features were used to track hand motions and detect six classes of hand patterns among the four regions of wheel, instrument cluster, gear shift and side rest, and the SVM with a linear kernel achieved over 80% normalized accuracy.

All mentioned ML-based driving behavior analysis studies reviewed in [22] detect different types of driving behaviors or activities and driving styles, and none concern accident detection. Another comprehensive study provided by Meiring et al. [4], reviews driver style analysis systems, the application of these systems and the underlying artificial intelligence algorithms applied to these applications. In this review, two major categories of driving study researches (i.e., driving style and applications assessing driver behavior studies) are presented. In the papers reviewed in [4], the most popular AI, ML and statistical algorithms are used on different driving research topics such as driver assistance, drowsiness detection, driver distraction detection, Eco driving, road and vehicle condition monitoring, fleet management, accident detection and insurance applications. Most of the mentioned literature in [4] focuses on detecting various driving styles such as normal and safe, aggressive, inattentive and drunk driving utilizing various data modalities such as multiple sensors (e.g., car networks, smartphone, telematics, video, etc.), but very few exist on accident detection. For instance, Lee et al. [29] proposed a log–linear model to predict crashes based on crash precursors (i.e., traffic flow conditions prior to the crash) using traffic flow data extracted from traffic loop detectors.

Bagdadi et al. [30] proposed the recognition of critical jerk on the naturalistic Virginia Tech Transportation Institute (VTTI) near-crash data with a detection rate of 86% on identifying safety critical braking events during car driving. Critical jerks [31] are defined as sudden changes in acceleration magnitude. Events involving an abrupt braking response, which creates a critical jerk, are classified as safety critical braking events.

### 2.2. Accident Detection

To more easily compare our study to past ones, we divide the related works from the literature into two categories: rule-based and ML-based accident detection studies.

#### 2.2.1. Rule-Based Accident Detection

Rule-based approaches are simple problem-solving techniques that are usually heuristic rules based on experts’ knowledge. These approaches are chosen to fit the given problem, and they work well only on specific data modalities and problems for which they are intended. Many research studies are carried out in the domain of rule-based accident detection systems based on traffic-monitoring data [32,33,34,35].

Traditional traffic accident prediction uses annual average traffic volume. Real-time traffic accident detection, on the other hand, uses monitoring devices such as induction loops, infrared detectors and cameras. Nevertheless, the usability of these specific devices is greatly limited by important installation and maintenance costs and their poor road network coverage which is typically restricted to well-known congestion zones, e.g., on highways, tunnels or bridges [11]. Conventional built-in automatic accident detection systems utilize impact sensors or the car airbag sensors to detect an accident and GPS to locate the accident location [36]. Sheu et al. [37] proposed a new methodology for real-time detection and characterization of freeway incidents. In this system, incident symptoms are identified from raw traffic data (the segment-wide inter-lane, intra-lane traffic dynamics, lane-changing fractions and queue lengths) utilizing the signal processing techniques, extended Kalman filtering and the modified sequential probability ratio test (MSPRT).

In addition to traffic data, smartphones are the next most used sensors in rule-based accident detection research. Zaldivar et al. [5] presented an application that, automatically informs emergency services about an accident by SMS based on the vehicle diagnostics interface (OBD-II). In this study, an accident is detected with airbag triggers on the basis of the force overload experienced in the event of a frontal collision. A slightly different approach proposed in [38] measures the tilt angle change using an accelerometer sensor and speed using GPS to detect the moment of collision and send an alert on detection of an accident. Another method proposed in [39] focuses on the use of the smartphone accelerometer to monitor vehicle speed and report an accident if it falls below a certain threshold. It can be noted however that smartphones may be unreliable for accident detection. The main issue with these systems is that the smartphone may tilt or fall any time inside the vehicle without a real accident, and thus, the probability of a false positive will increase and a false alarm will be reported. Past literature ubiquitous computing has shown that there are also significant differences between smartphone brands: a machine learning model trained for a smartphone might see a strong degradation in its performances when used with another smartphone of a different brand, even if the latter contains the same sensors [40].

#### 2.2.2. ML-Based Accident Detection

ML techniques notably differ from an algorithmic point of view depending on what type of input data is used to apply them. Due to the large data requirements of ML and relative scarcity of labeled car accident data sets, only a few related studies can be found. Among them, three main sensor modalities can mostly be seen combined with ML in the literature: traffic data, sensor data and car internal signals.

Studies that propose accident detection based on ML approaches are mostly based on traffic monitoring data. For instance, Ozbayoglu et al. [41] used Istanbul City traffic-flow data for the year 2015 from located RTMS (real-time monitoring system) sensors. Specific handcrafted features characterizing lane velocity, occupancy and capacity usage were manually crafted and computed for each sensor. The extracted features were then fed into a nearest neighbor model, a regression tree and a feed–forward neural network model to predict the possibility of an accident. The overall accuracy of their models are mostly over 99%, but with a considerable number of false alarms.

A review of the literature related to IoT (Internet of Things)-based accident detection, prevention and reporting systems is provided by Alvi et al. [13]. In this review various applications of IoT are introduced and referenced. Accident detection papers in the study are grouped into two categories: (1) conventional- and (2) ML/AI-based accident detection techniques. For the second category of the reviewed accident detection papers in [13], three key ML/AI-based accident detection approaches are discussed: (a) fuzzy logic, (b) SVM and (c) artificial neural network (ANN). Pan et al. [42] used vehicle speed, acceleration and lane changing factor from a microscopic traffic simulator and classified incident vs. non-incident with a SVM. With their proposed methodology, the SVM accuracy based on speed data was almost 100%. In their work, they imply the assumption that every vehicle collects its own traffic data and sends them through an OBU (on-board unit). Then the traffic data is collected via RSU (roadside units) and uploaded to the central service for processing. The simulated scenarios they tested only consider accidents on three-lane urban roads at traffic lights.

Harlow et al. [32] proposed a system that involves a method for processing and recognizing accidents from recorded vehicle acoustic signals at intersections and construction sites. A database of vehicle sounds, car braking sounds, construction sounds and traffic sounds was created. The mel-frequency cepstral coefficients were computed as a feature vector given as input in a neural network classifying between crash and non-crash events. In their study the classification testing results achieved 99% accuracy. Although their proposed methodology is highly efficient, this system is bound to specific locations e.g., intersections and construction sites.

Other works have attempted to leverage data coming from external sensors (video, audio, movement) for accident detection. An accident detection approach based on a convolutional neural network (CNN) was proposed by Ghosh et al. [43] using video footage obtained from the CCTV cameras installed on highways. By placing CCTV cameras and a Raspberry Pi 3, using a pre-trained CNN model from 10,000 accident frames and 10,000 non-accident frames, their model can detect accidents to an accuracy of about 95%. The main disadvantage of such models is that accidents outside the area covered by cameras are overlooked. Monitoring all roads and highways with Pi cameras and Raspberry Pi is a costly approach for accident detection. Maintaining privacy and security is another disadvantage of the proposed system. Finally, weather can affect the cameras viewability.

The abovementioned works present approaches utilizing systems that are limited by challenges such as maintaining the privacy and security of users, huge expenses of a large-scale network, many manufacturers, industries, power consumption, architecture challenges, heterogeneity, mobility and interoperability problems. A similar solution to ours using internal car data to bypass the drawbacks of other monitoring devices was proposed by Osman et al. [44]. This study introduces a ML model to predict collisions on the SHRP2 NDS vehicle kinematic data (speed, longitudinal acceleration, lateral acceleration, yaw rate and pedal position). They hypothesize that vehicles experience micro-level turbulence in their kinematic patterns over a period of time known as the turbulence horizon prior to a crash. They use the input feature as the standard deviation of the vehicle kinematics parameters in the period starting at the beginning of the turbulence horizon and ending at the beginning of the prediction horizon (the time interval between the predicted hazard time and the impact time) as features in their study. To classify near crash data and normal driving data, several classification algorithms are trained and compared. Although this model achieved a substantially high 99%F1 score, the feature extraction approach used in this study was based on label re-computation, which might not guarantee features generic enough to be successfully re-applied on data other than SHRP2 NDS.

Two major points distinguish our study from [44]: First, our work focuses on accident detection rather than prevention. Therefore, we do not include near-crash scenarios in our data set. Second, and more importantly, our work is also testing feature-learning approaches based on deep neural networks that learn features in an automated way without the need of any prior knowledge.

## 3. Materials and Methods

This section introduces the proposed ML framework (Figure 1) and describes in details how to use it for car accident detection based on internal car data. Since the largest improvements in classification performances can usually be obtained at the feature extraction stage, we use our proposed framework to analyze and compare the respective performances of state-of-the-art feature extraction techniques. To do so, we fix the acquisition, pre-processing and classification steps as described in the following subsections.

### 3.1. Data Acquisition

Data acquisition is the process of defining an experimental protocol to properly set up sensors, defining a strategy to properly annotate the data with labels, sampling relevant sensor signals, converting the resulting samples into digital numeric values and acquiring and merging the data from appropriate sources. Access to a suitable data set and the quality of the data are the basic prerequisites for the successful establishment of ML-based studies. Since our study focuses on accident detection based on real data, finding a suitable labeled database was necessary in advance. In the frame of our experiments, it was decided to use the SHRP2 NDS [45] database acquired from Virginia Tech Transportation Institute (VTTI) [10]. In the following section, a brief description of the SHRP2 data set is presented.

#### SHRP2 Data Set

The SHRP2 research project in the study of naturalistic driving behavior, monitored around 3400 apprentice drivers from over 277 unique car makes/models from six locations across the United States. The participants’ vehicles were equipped with a data acquisition system (DAS) that included a forward radar, four video cameras, a front-facing wide-angle camera, accelerometers, vehicle network information, a geographic positioning system, on-board computer lane tracking, various computer vision algorithms and additional data storage capabilities. An accident data set (mostly from cars, followed by SUV-crossovers, pickup trucks and van–minivans) was created from 5,512,900 trip log files extracted from the SHRP2 naturalistic driving study (NDS). To identify accidents, a team of data analysts and data quality coordinators manually validated and analyzed the log files to annotate the data. An accident was defined as any contact a vehicle has with an object, either moving or stationary, at any speed in which kinetic energy is measurably transferred or dissipated. The definition also includes non-premeditated departures from the road where at least one tire leaves the paved or intended route road surface. The acquired data set consists of 546 synchronized accident events containing the time-series data channels shown in Table 1.

### 3.2. Data Pre-Processing and Segmentation

Data pre-processing refers to operations applied to clean the data of flaws usually caused by data transmission errors or sensor failures. They include, in particular, operations such as eliminating any duplicates, irregularities in the data, normalizing the data to compare, filling out missing data values, which is a commonly encountered problem, and providing the ML model data that are consistent to improve the accuracy of the obtained result. Usually, original raw data records are too long and might not contain homogeneous information. Therefore, splitting the data into shorter segments is needed. Segmentation is the procedure of splitting the signal values (in our case time series) into separate time intervals called windows, depending on the attribute or the behavior of the desired classes. Figure 2 illustrates the pre-processing and segmentation steps applied on the SHRP2 data set in this study.

First, to synchronize the sensor channels of the SHRP2 data set, we re-sampled data from different sensors at a frequency of 40 Hz. Afterwards, to bring the sensor values into the same range, min–max scaling normalization was applied. For the segmentation, a traditional sliding time window approach was employed. The labeling was performed by using the information from event descriptions provided with the SHRP2 accident data set, which contains the event start and end time stamps. Based on this information, we decided to set the segment length to T=100 which corresponds to approximately 2.5 s. This duration was chosen to be equal to the maximum event duration based on the SHRP2 data description. An overlapping factor of 50% was also chosen to segment the time series.

The labeling process adds a label of zero or one to each time window depending on whether the frame at a specific time stamp is part of the event. Windows containing more than half of the duration of the frame dedicated to an accident event are labeled as one and the rest of the windows as zero. After the aforementioned steps, our data consists of 34,339 time windows sized of shape T×S where T=100 is the length of the time window, and S=4 is the number of sensor channels. Among these 34,339 time windows, only 2281 are positive samples. Based on these numbers, as expected, we are dealing with a notably imbalanced data set in which the negative class represents 93% of the whole data set.

### 3.3. Feature Extraction

Feature extraction is the process of reducing the dimension of the raw data into an informative abstract representation of the classifier. Extracting features reduces the complexity of training a classifier by reducing the size of its input data and getting rid of information irrelevant for the classification problem to be solved.

In this section, the feature extraction techniques used in our study are briefly explained. We decided to work with the most common state-of-the-art feature extraction approaches in time-series analysis and apply them on four in-car signals, including gas pedal position, speed, steering wheel position and acceleration.

Feature extraction methods from both feature engineering and feature learning are presented in this paper. The most commonly used feature extraction approach is the traditional feature engineering method consisting of handcrafted features that we refer to as HC in the following sections. HC features have been implemented for decades and still serve as a powerful tool when combined with ML classifiers. Traditionally, HC features are engineered based on knowledge expertise of the data which is not always available. In this case, it is common to use simple statistical attributes computed on the time series, which have shown to perform well in practice despite their simplicity [46]. Feature engineering based on heuristic rules found in the past literature such as [15,16] was also adapted and tested in our experiments, though due to insufficient performance the method and results are not presented in this paper.

The second category of feature extraction approaches is deep feature learning. Deep feature learning refers to the automated learning of features using deep neural networks (DNNs). An ANN is a composition of *L* parametric functions represented by “layers”. ANNs consist of 3 layers: input, hidden and output. A deep neural network (DNN) is an ANN with at least two hidden layers. Each layer consists of a group of multiple neurons. Neurons are simple non-linear computational units that output a single value given several inputs. In the general case, the layer $l_{i}$ with *i*∈ 1 … *L* takes as input the output of the previous layer $l_{i-1}$ and applies a non-linearity to compute its own output. The last layer usually provides an estimation of class probabilities associated with the data given as input to the model. For this purpose, a softmax activation layer with a number of neurons equal to the number of classes is traditionally used. DNNs have shown to be extremely good feature learners, especially in image classification [17].

Since our evaluation framework fixes the classifier, an approach similar to what was done by Li et al. [14] and Girshick et al. [47] was employed to use DNNs as feature extractors. The DNNs were first trained in a supervised way using a softmax layer. The latter was then removed to let the DNN output feature vectors that were used train the classifier fixed in our framework. A brief explanation of the abovementioned feature extraction methods chosen for our study follows.

#### 3.3.1. Feature Extraction Based on Handcrafted Features (HC)

Traditional HC feature extraction has been implemented for decades, and due to its simple set up, it still serves as a powerful baseline when combined with ML classifiers. The HC features extract information from simple statistical attributes such as minimum, maximum or a percentile or more elaborated descriptors such as features related to the frequency–domain based on the Fourier transform of signals. For HC features, we computed simple statistical features that are commonly used in many application domains using time-series data [14,46]. These features consist of 18 values computed either on the time series or their power spectrum as shown in Table 2. These features are computed on each sensor channel individually. The features from all channels are then concatenated together to form a feature vector of size 18×4=72.

To improve the classification efficiency, prevent overfitting (less opportunity to make decision based on noise) and provide better insight into relevant features, in our paper HC feature extraction is followed by a feature selection approach. Three popular feature selection protocols, belonging either to the family of filter-based or wrapper-based methods, were tested in our study: ReliefF and Fisher score from filter-based and recursive feature elimination (RFE) from wrapper-based feature selection approaches [8,48,49,50]. Of these three methods, only RFE could yield proper improvements in performance. Therefore, only its results are reported in our paper.

RFE feature selection is a wrapper-based approach that aims at evaluating different sets of features by training and evaluating a classifier for each set and comparing their classification performances. Each wrapper approach proposes a strategy to select the sets of input features to test and avoid having to test all configurations, which would be too computationally expensive. RFE starts with using all features as a whole subset and training the classifier. Then it eliminates features by recursively considering smaller and smaller sets of features. In our study, features were eliminated based on feature importance scores returned by the classifier we chose (random forest and support vector machine). The n= step size and number of features are based on the lowest score eliminated. The procedure is recursively repeated until the desired number of remaining features or level of performance is reached.

#### 3.3.2. Feature Learning Based on Multi-Layer-Perceptron (MLP)

An MLP constitutes the simplest and most traditional architecture for deep learning models. This form of architecture is also known as a fully connected network since the neurons in layer $l_{i}$ are connected to every neuron in layer $l_{i-1}$ with *i*∈[2,L], *L* being the number of layers. Parameters referred to as weights are attributed to each connection. Consequently, the connection between two consecutive layers of a MLP can be represented by the following equation:(1)$$x_{l_{i}}=f\left(W_{l_{i}}*x_{l_{i-1}}+b_{l_{i}}\right)$$

With $n^{\left(i\right)}\in N^{*}$ number of neurons in layer $l_{i}$, $x_{l_{i}}\in R^{n^{\left(i\right)}}$, $W_{l_{i}}\in R^{n^{\left(i\right)}\times n^{\left(i-1\right)}}$, $b_{l_{i}}\in R^{n^{\left(i\right)}}$, $W_{l_{i}}$ being the matrix of weights connecting the neurons of layer $l_{i-1}$ to layer $l_{i}$, $b_{l_{i}}$ the vector of biases in layer $l_{i}$ and $x_{l_{i}}$ the output of layer $l_{i}$. Figure 3 illustrates the schematic of the MLP network used in our study.

#### 3.3.3. Feature Learning Based on Convolutional Neural Networks (CNN)

CNNs mainly contain convolution layers and pooling layers, and some deep learning architectures include batch normalization layers (see Figure 4). They have been successfully applied in image recognition [51,52], in various natural language processing tasks [53,54] and in timeseries analysis [55]. The general form of using a convolution for the centered time stamp t is given in the following equation:(2)$$\forall t\in \left[1,T\right],c_{t}=f\left(\omega *x_{t-l/2:t+l/2}+b\right)$$
where ∗ designates the convolution product, $c_{t}$ the result of the convolution at time *t*, *f* the activation function, x a 1D input, ω the convolutional filter of length *l* and b a bias parameter. A convolution can be seen as applying and sliding a filter over a time series, or in other words as a generic non-linear transformation of an input vector x. For instance, if convoluting a time series with a filter of length 3 with values equal to $\left[\frac{1}{3},\frac{1}{3},\frac{1}{3}\right]$ is the equivalent of applying a moving average with a sliding window of size 3.

The convolutional maps obtained after applying several convolutional kernels are usually given as input of a pooling layer, which can be either local or global. A local pooling operation, such as averaging or taking the maximum value in a sliding window is applied to downsample the input of the layer. For global pooling, the downsampling operation is applied across the entire input time dimension, resulting in a single output value. A normalization layer is sometimes added to the network to help the network converge quickly during training. A batch normalization [56] and instance normalization [57] are two popular normalization layers used in CNNs for time-series analysis. When used for classification, CNNs are usually connected to a classification MLP, whose last layer is a softmax layer, giving a distribution score over the class variables in the data set.

#### 3.3.4. Feature Learning Based on Long Short-Term Memory (LSTM)

Long short-term memory networks are a type of recurrent neural network (RNN) being a special type of DNN that use loops and connections between nodes along the temporal sequences to save dynamic temporal information and carry it along the network. LSTM cells introduce internal mechanisms called gates that can regulate the flow of information over time by storing it in an internal memory and update output or erase this internal state depending on their input and the state at the previous time step [58]. Gate operations can be described as follows:(3)$$\begin{array}{c} f_{t}=\sigma \left(W_{f}\cdot \left[h_{t-1},x_{t}\right]+b_{f}\right) \end{array}$$(4)$$\begin{array}{c} i_{t}=\sigma \left(W_{i}\cdot \left[h_{t-1},x_{t}\right]+b_{i}\right) \end{array}$$(5)$$\begin{array}{c} o_{t}=\sigma \left(W_{o}\cdot \left[h_{t-1},x_{t}\right]+b_{o}\right) \end{array}$$(6)$$\begin{array}{c} m_{t}=f_{t}⊗m_{t-1}+i_{t}⊗\sigma \left(W_{c}\cdot \left[h_{t-1},x_{t}\right]+b_{C}\right) \end{array}$$
where $x_{t}$ is the input vector to the LSTM cell and $h_{t-1}$ the hidden state vector also known as the output vector of the LSTM cell. $f_{t}$, $i_{t}$ and $o_{t}$ represent forget, input and output gates, respectively. These gates have their own weights ($W_{*}$), bias ($b_{*}$) and activation functions (σ). The functionality of an input, output and forget gates are used to block the input of the cell, block its output, and erase its internal memory at time *t*. $m_{t}$ is the memory state of the cell at time *t*, and ⊗ is the element-wise multiplication of two vectors. The architecture of the RNN with LSTM layers containing LSTM cells is shown in Figure 5. Like all DNNs, there are different architectural variations of LSTM layers, and the last layer is followed by dense and softmax layers.

#### 3.3.5. Feature Learning Based on an Autoencoder (AE)

Autoencoders are special types of DNNs that are trained to reproduce input data on their output using a loss function such as mean squared errors in an unsupervised manner. An AE applies a dimensionality reduction by first projecting input data into an embedding space (of usually smaller dimensionality than the one of the input space) using an encoder to then decompress the embedding to match the original input as closely as possible using a decoder. By construction, an AE also always has the same number of inputs as outputs. Figure 6 shows a schematic view of the architecture of an AE. For feature extraction, an AE is first trained in an unsupervised way to reconstruct its inputs on its output layer. Then the decoder is removed, and the encoder is used to output feature vectors.

### 3.4. Classification

Classification is the final step of the ML framework which trains a model to predict class labels (categories) given an input feature vector associated with a specific data segment. The classifier constructs a separation between different classes in the feature space. Support vector machine (SVM) [59], random forest (RF) [60], k-nearest neighbors (KNN) [61], decision tree [62], etc., are among the most popular classifiers used in the past literature. In this paper, a soft-margin SVM (known as a C-SVM) with a radial basis function (RBF) kernel and a RF classifier are the two chosen classifiers due to their high performance and ability to overcome overfitting in the case of high dimensional data [14,63,64].

A SVM has been shown to be a very effective linear classifier that can also effectively be applied to non-linear cases with the kernel trick. C-SVM is a variation of a SVM that allows misclassifications during the training to reduce overfitting by regulating the soft-margin parameter *C* that controls how much misclassification is allowed. When training a SVM with the RBF kernel, two parameters must be considered: *C* and gamma. The parameter *C* trades off misclassification of training examples against simplicity of the decision surface. Proper choice of *C* and gamma is critical to the performance of the SVM. A grid search was performed to find the optimal *C* and gamma values for each feature set extracted by the abovementioned methods. The RF is a popular classifier that uses bagging mechanisms with decision trees to reduce their propensity to overfit. The main hyper-parameter of RF is *T*, the number of trees. Similar to SVM parameters, the number of RF trees *T* was optimized by a 1-D grid search for each feature set separately.

It should be noted that DNNs are trained directly with a classification layer. To apply the aforementioned classifiers on the DNNs features, the following procedure needed to be done. At first the model was trained as usual, and then the classification layer (softmax) was removed to assign the output of the penultimate layer as an input feature to the classifier. In feature extraction based on AE, the output of the encoder is the assigned feature after training the full AE model.

## 4. Experiments and Results

This section is about the implementation details of all applied algorithms, the evaluation setting and the results. All the implementations in this study were coded in Python Feature learning approaches using DNNs were implemented using the *Keras 2.1.0* framework with a *Tensorflow 1.14.0* [65] backend, *scikit-learn 0.21.3* [66] and trained using the ADADELTA optimizer [67] with default parameters (initial learning rate of one) for 50 epochs with a batch size ranging from 100 to 1000.

To have a comprehensive measure of our model’s performance, throughout the whole data set, a K-fold cross-validation [68] was applied to the data set. The number of folds in this case was set to K=5 and for each training run, we selected a single partition from our five folds to be the test set and used the rest for training. The details of the experiments for each feature extraction algorithm are described as follows:**HC:**    
The HC features consisted of 15 statistical values directly computed on the time series, and 3 frequency-related on their power spectrum were computed on each sensor individually and concatenated together. Then RFE feature selection with an elimination size of three was applied.**MLP:** 
The MLP architecture used in this study contained three dense layers and REctified Linear Units (RELU) activation. MLP usually takes 1D inputs only, therefore a flattened layer was used to convert the 2D input to 1D. According to the recommendations of [14,56], a batch normalization layer was placed directly after the network input to improve results. Three fully connected dense layers with RELU activation function, containing 2000 neurons each, and a final softmax layer built up the MLP network used for our study (see Table 3). In Table 3, the values for the hyper-parameters used for feature learning approaches in this study are shown. Optimizing the hyper-parameters of DNNs is an important and difficult topic. Optimal parameters for our models were chosen after testing several manually selected configurations. Manual hyper-parameter selection is the default approach in the literature due to the absence of other more elaborated high performing approaches.**CNN:** 
As listed in Table 3, the CNN layout consisted of three blocks of batch normalization and a convolutional layer with RELU activation followed by dense and pooling layers. The CNN design was based on [14] with some modifications, including a reduction in the size of the convolutional kernels and an increase pooling window size, while keeping the amount of kernels the same for each block.**LSTM:** 
The values of all hyper-parameters for the LSTM architecture are provided in Table 3. Like other ANNs used in this paper, a batch normalization layer was added at the beginning and a dense and softmax layers at the end of the network. The gate activation used in the LSTM cells is a sigmond function, and in the dense layers, a tangent activation function was used.**AE:**    
The AE architecture consisted of simple dense layers (three dense layers for the Encoder and then three for the Decoder designed as a mirror), with ReLU for the activation function. Different numbers of dense layers were tested, and the one achieving the best performance is presented in Table 3.

**Table 3 sensors-22-03634-t003:** Hyper-parameters of the ANN models on the SHRP2 data set.

Model	Parameter	Value/ Type
MLP	. # Dense layers	3
	. # Neurons in each layer	2000
	. Activation function	ReLU
CNN	. # Conv. blocks	3
	. Conv. kernel size for blocks 1, 2 and 3	(5, 1), (4, 1), (3, 1)
	. # Conv.kernels in each block	50
	. Pool size for blocks 1, 2 and 3	(2, 1), (3, 1), (4, 1)
	. # Neurons in the dense layer	1000
	. Activation function for the Conv. blocks	Tanh
	. Activation function for the dense layer	ReLU
LSTM	. # LSTM layers	2
	. # Output dimensions for each LSTM cell	600
	. # Neurons in the dense layer	512
	. Activation function for the dense layer	ReLU
AE	. # Encoder dense layers	3
	. # Neurons in layers 1, 2 and 3	5000, 3000, 1000
	. Activation function	ReLU

The evaluation of our feature extraction framework is based on three different metrics, average F1 score, overall accuracy and weighted F1 score. F1 score is the harmonic mean of precision and recall, and average F1 score is the mean value of each class F1 scores. As mentioned in Section 3.1, our data set was strongly imbalanced with only 6.65% positive samples, and both accuracy and weighted F1 score are very biased in case of an unbalanced data set. For this reason, the average F1 score, which is the mean value of each class F1 score, is considered as our main evaluation metric due to its ability to take class imbalance into account [69].

The results of the aforementioned feature extraction approaches with both SVM and RF classification methods are provided in Table 4 and Table 5, respectively.

Three main observations can be drawn from Table 4 and Table 5. First, the choice of the classifier impacts the final classification performance. RF improved the average F1 score of HC feature extraction to 71.78%, and a notable improvement of the RF with almost 10% more for average F1 score is for LSTM feature learning. Second, deep feature learning outperforms feature engineering. In particular, CNN surpasses other methods by a remarkable average F1 score of 79.10% and 78.39% for SVM and RF, respectively. Finally, the quite decent performances with unsupervised deep feature learning (AE) when compared to supervised feature learning, is a remarkable observation of this study.

## 5. Analysis

The experiment on applying feature extraction approaches on a SHRP2 crash data set reveals the following points: First, all feature learning approaches except LSTM with both classifiers outperform the feature engineering. LSTMs are usually prone to high computation time and are difficult to tune. Another reason for the poor performances of LSTM is the length of the time horizon (T=100) of the input samples being too long. This happens quite often when dealing with time series and is one of the reasons CNNs are preferred over RNNs for time-series processing in the literature.

CNN feature learning obtains a stable top performance with both classifiers. Obtained results are consistent with the literature, which seems to indicate that CNNs are the most reliable architecture for time-series classification [70,71]. To understand the difference between HC features and features created by deep feature learning, the best features extracted by both approaches were analyzed in our paper. For HC, it is possible to use the ranking of the RFE algorithm which was applied for feature selection.

Figure 7 shows the RFE on HC features for the five cross-validation folds. In these figures, the x-axis is the RFE eliminating steps, and the y-axis shows eighteen HC features of each sensor channel. The RFE method removes features iteratively, starting with the ones with the least impact on the final classification performance. As can be seen in all five figures, the HC features extracted by gas and acceleration sensors are the ones that are eliminated last, meaning that features from these two sensor channels are the most important HC features selected by RFE.

For DNNs, finding the best features is more challenging since features learned by DNNs are hard to interpret. Instead, we decided to use an approach based on the Jacobian matrix of the model to determine which sensor channels are the most important and whether this matches the observation of the HC results. In case of considering a trained DNN as an approximation of a multi-input/multi-output function $f:R^{L\times S}\to R^{C}$, where *L* is the length of a multichannel segment X belonging to the target data set $X_{T}$, *S* is the number of channels, and $C_{T}$ is the number of classes, it is possible to compute the Jacobian matrix of function f. Each Jacobian value $J_{c,l,s}\left(X\right)$ in the Jacobian Matrix, represents the importance of $x_{ls}$—the value at the *l*th time point (1≤l≤L) of the *s*th sensor—on the predictive function for the *c*th class. It can therefore be used to indicate which parts of the input have the most impact on the output classification score. With this, it is possible to propose a ’Jacobian score’ similar to [72] that would indicate how important to the final classification score a specific sensor channel is.

A channel-wise Jacobian score for $\;\omega_{s}\left(X\right)$ as the average of absolute $J_{c,l,s}\left(X\right)$ over all the *L* time points and all the $C_{T}$ classes is,
(7)$$\begin{array}{c} \omega_{s}\left(X\right)=\frac{1}{C_{T}}\frac{1}{L}\sum_{c=1}^{C_{T}}\sum_{l=1}^{L}|J_{c,l,s}\left(X\right)| \end{array}$$

In addition, as described in [72], the global channel-wise Jacobian score $\Omega_{s}$ is averaging $\omega_{s}\left(X\right)$ over all examples $X_{T}$ in the data set, where $\left|(X_{T})\right|$ refers to the cardinality of $X_{T}$:(8)$$\begin{array}{c} \Omega_{s}=\frac{1}{\left|(X_{T})\right|}\sum_{X\in X_{T}}\omega_{s}\left(X\right) \end{array}$$

A high $\Omega_{s}$ indicates a high importance of the sensor channel for the classification problem. Figure 8 shows the Jacobian scores for the four input sensor channels (speed, gas position, steering angle, acceleration) and for each of the five CNNs trained on each cross-validation fold. According to the Jacobian scores of these sensors, steering-angle has the highest impact on the result. Speed and acceleration are the second most important input signals. Finally, this study shows that gas-position was consistently found to be considered as the least useful channel by CNN, which is opposite to feature extraction with HC that obtained its best features from this channel. This would indicate that the features learned by HC and CNN are different in nature and might be complementary. Additionally, we present a hypothesis regarding why steering-angle information was regarded as useful for the classification problem by CNNs and not by HC. The steering-angle signal, compared to the other sensor channels, is characterized by strong and rapid fluctuations that might contain valuable frequency-based information for the considered classification problem. Contrary to CNN, this information might not have been captured well enough by the HC features that were extracted.

Analysis of the feature importance results led us to combine CNN features with the optimal ranked HC features to improve the results. The same steps were used to calculate the features as described in Section 3.3.1. Then the RFE was used to rank all features and keep only the three best ones. These features were then appended to the 1000 features of CNN in Section 3.3.3. Combined CNN features improved the average F1 score from 79.10% to 80.12% for a SVM and from 78.39% to 79.10% for RF classifier which seems to confirm our hypothesis that both features contain complementary information.

## 6. Conclusions

In this paper, we presented a framework for accident detection from the most basic sensors in cars in the SHRP2 naturalistic data set using machine learning approaches. This framework was used to perform a study testing various feature extraction and classification approaches. The state-of-the-art feature extraction methods, including traditional manual feature extraction and feature learning, were combined with two classifiers. CNN features with a SVM classifier outperformed all other tested approaches including HC with an accuracy of 85.72%, an 84.9% weighted F1 score and an 79.10% average F1 score. This result is very promising, considering that the data used in this study are based on naturalistic accidents, and very few samples are severe accidents, recognizable by only four basic in-car sensors. Additionally, interpretability studies showed that the HC and DNN approaches were extracting their optimal features from different sensor channels and could therefore effectively be combined because of their complementarity.

The first major limitation of our study is that we use a single data set due to the extreme scarcity of data for this application, which limits the generalization capacity of our findings. Second, it is difficult to compare our findings to other studies due to different accident definitions and different types of input sensor channels.

Based on our observations and study conclusion, the *lessons learned* are listed briefly as follows:It is possible to obtain promising results with ML for the detection of accidents using basic in-car sensor data.A deep learning feature extraction method performs better in comparison with HC, and unsupervised feature extraction remarkably achieves the second best performance score.

Therefore, our future work will include four main points: First, the number of sensor channels tested in this work is limited. We would like to explore other combination of channels as well as other learning algorithms to further improve our classification performance. Including more sensor modalities such as lateral acceleration and yaw rate to the data set which are still counting as initial sensors and are available in all cars and analyzing their influences on the detection performance is a potential study investigation. The second possible way to improve the accident detection results is to employ more advanced feature selection algorithms such as embedded feature selection methods for unbalanced data sets (e.g., [73]) as future work. Third, following the promising performances of unsupervised deep feature learning (AE), further investigation of unsupervised feature learning techniques for future work is recommended. Finally, to bypass the struggle of insufficient labeled data, one possible solution would be to investigate transfer learning [74].

## Figures and Tables

**Figure 1 sensors-22-03634-f001:**
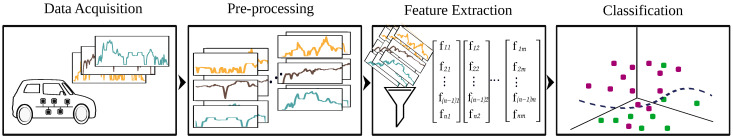
Machine learning framework for accident detection. First, time-series data is being acquired from in-car network signals. After pre-processing and segmentation, different feature extraction approaches are applied and compared. Finally, classifiers are trained and tested on the extracted features.

**Figure 2 sensors-22-03634-f002:**
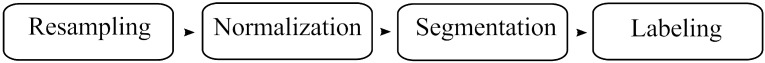
Pre-processing procedure applied on SHRP2 data set for accident detection.

**Figure 3 sensors-22-03634-f003:**
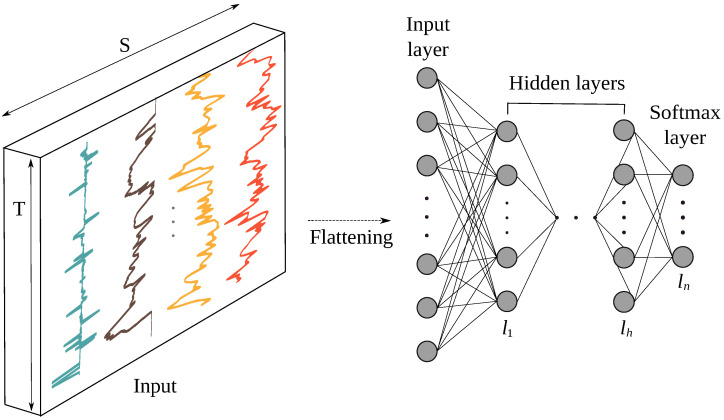
Architecture of a MLP model for accident detection with *h* hidden layers, *n* number of classes and *S* number of sensor channels. Input data are first flattened into a (T×S)-dimensional vector and and then fed to the hidden layers. All layers are fully connected.

**Figure 4 sensors-22-03634-f004:**
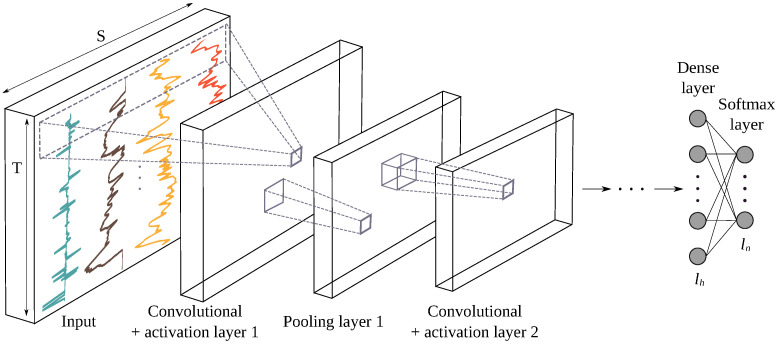
Architecture of a CNN model for accident detection. The designations *n*, *h* and *S* are the number of classes, layers and sensor channels, respectively. Convolutional layers apply convolution products on all convolution maps of the previous layer. Pooling layers then downsample the convolutional output and pass it to the next convolutional layer.

**Figure 5 sensors-22-03634-f005:**
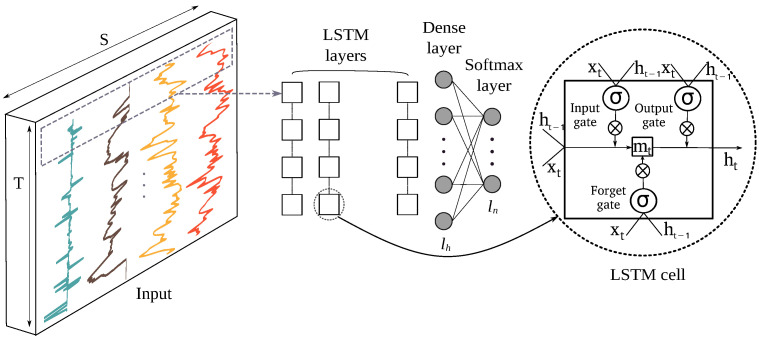
Architecture of a LSTM network for accident detection. The designations *n*, *h* and *S* are the number of classes, layers and sensor channels, respectively. The cells in the LSTM layers have one input, forget and output gates. $x_{t}$, $m_{t}$ and $h_{t}$ refer to the cell input, memory, and output at time *t*, respectively, and σ designates the activation function.

**Figure 6 sensors-22-03634-f006:**
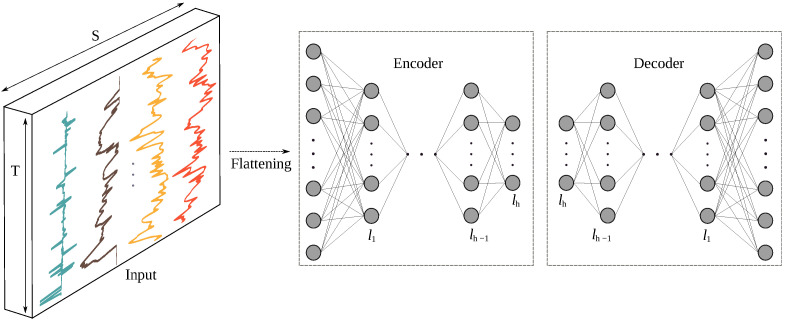
Autoencoder’s architecture used for accident detection consists of encoder and decoder sections. The encoded layer is used as features for the feature learning purposes; *S* is the number of sensor channels and *h* denotes the number of hidden layers in both the encoder and decoder.

**Figure 7 sensors-22-03634-f007:**
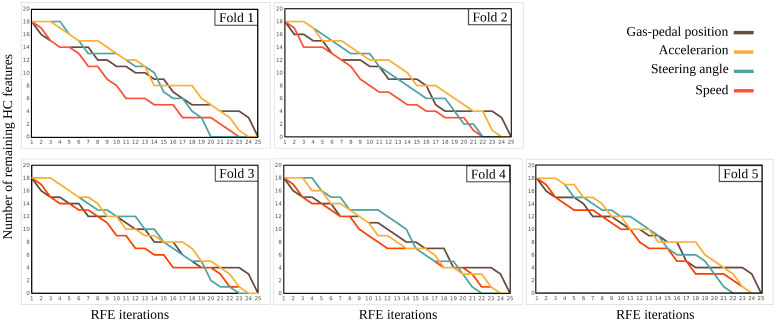
RFE results per sensor channel for five cross-validation folds. The y-axis shows the number of remaining features from each sensor, and the x-axis shows the step by step removal of features from each sensor.

**Figure 8 sensors-22-03634-f008:**
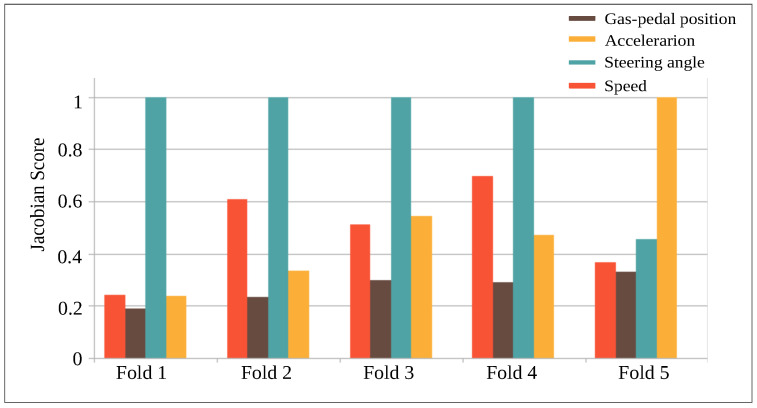
Jacobian score for CNN of five cross-validation folds of four signals. A higher Jacobian score indicates that the sensor channel contributed to the learning of more discriminative features. The y axis represents the normalized magnitude of the Jacobian score for each sensor channel.

**Table 1 sensors-22-03634-t001:** SHRP2 data set sensor channels.

Variable Name	Unit	Description
Time stamp	millisecond	Time since beginning of trip, in milliseconds
Gas pedal position	none	Position of the accelerator pedal
		collected from the vehicle network
		and normalized using manufacturer specs
Speed network	km/h	Vehicle speed indicated on
		speedometer collected from network
Steering wheel position	degree	Angular position and direction of
		the steering wheel from neutral position

**Table 2 sensors-22-03634-t002:** List of the handcrafted features used in our study. Each feature is computed on each sensor channel independently.

Handcrafted Features		
Maximum	Average	Auto-correlation
Minimum	Skewness	First-order mean
Percentile 20	Kurtosis	Second-order mean
Percentile 50	Interquartile	Standard-deviation
Percentile 80	Zero-crossing	Norm of the first-order mean
Spectral entropy	Spectral energy	Norm of the second-order mean

**Table 4 sensors-22-03634-t004:** SVM classification evaluation metrics (in percent) of different tested feature extraction models on the SHRP2 data set.

Methods	Accuracy	Weighted F1 Score	Average F1 Score
HC	94.34	92.99	66.56
MLP	83.60	82.30	75.00
CNN	85.72	84.9	79.10
LSTM	76.81	72.01	57.90
AE	83.40	82.40	75.50

**Table 5 sensors-22-03634-t005:** RF classification performance metrics (in percent) of different feature extraction models of five cross on the SHRP2 data set.

Methods	Accuracy	Weighted F1 Score	Average F1 Score
HC	94.97	93.95	71.78
MLP	84.06	83.57	77.47
CNN	85.72	84.19	78.39
LSTM	78.00	76.61	67.22
AE	84.22	83.74	77.67

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
