# Peer review of "A Machine Learning Framework for Automated Accident Detection Based on Multimodal Sensors in Cars"

_sensors, 2022, doi:10.3390/s22103634_

Round 1

Reviewer 1 Report

The paper focuses on the use of different machine learning approaches to highlight the most important naturalistic driving-related parameters for road accident detection tasks, as well as to define a high efficiency model that can aid in identifying such an outcome at its occurrence. The manuscript is written in good English and is scientifically worthy of interest, rigorous in presenting the employed method and the obtained results. The following indications are provided to the authors for increasing the soundness of the paper (only page number is provided because of the lack of line numbering):

Introduction

Page 1/21 – “…the number of traffic related fatalities continues to rise steadily” is not a fact that only depends on the proper detection of the accidents, but also on the proper, fast recognition of its severity and the related injury risk for the occupants. A large number of studies is available in the literature that consider the need to also employ injury risk models to enhance accident detection algorithms, e.g., 10.1080/15389588.2014.927577, https://doi.org/10.1016/j.aap.2020.105864, 10.1016/j.aap.2016.09.028, the latter highlighting some figures of potential reduction in severe and fatal injuries obtainable by the use of smart emergency call devices. The authors are invited to also address and discuss these elements for completeness.

Results

Page 18/21 – “Analysis of the feature importance results led us to combine CNN features with the optimal ranked HC features to improve the results”: the real problem of the considered classification methods is that the indication regarding the effect of the different features depends on the results of an a posteriori calculation: once the best possible model is obtained for the specific problem, the relevant variables are ranked considering their effect on the outcome variable taking into consideration the formulation of the derived model. However, there are other techniques as ReliefF that allow users to a priori rank the variables that most prominently affect the phenomenon to be studied, without initially employ an a posteriori ranking method as the random forest; this is not a novelty even in the accident analysis field, as demonstrated by the study https://doi.org/10.1016/j.aap.2020.105864. The authors have undoubtedly provided good results as well as theoretical bases for their validity even without considering ReliefF-like algorithms, so no additional processing needs to be applied on the data; however, this could be an interesting point to discuss in the manuscript also in terms of future steps of the research.

Typos

Page 4/21 – “discuses” instead of “discusses”

Page 4/21 – “to explores” instead of “to explore”

Page 4/21 – “Critical jerks [24], are defined” instead of “Critical jerks [24] are defined”

Page 13/21 – “associated to” instead of “associated with”

Reviewer 2 Report

Title: A Machine Learning Framework for Automated Accident Detection based on Multimodal Sensors in Cars

Status: Minor revision

Introduction

  • In the third paragraph it is mentioned that not much has been done in the field of accident detection using Machine Learning, but at the beginning of the fourth paragraph the idea is taken up again indicating that accident detection has been explored in the past literature on Machine Learning. The wording of these two parts must be changed so that the idea expressed does not seem contradictory.
  • At the end of page 2 it is mentioned that accident detection based on Machine Learning can be categorized depending on the type of data used to train the model and the categories are identified: traffic data and external sensors. It does not delve into the reasons why it is difficult to access traffic data nor the justification for why external sensors can be difficult to configure and unreliable. Do these disadvantages apply to all sensors? Which have been the most used?. Explaining the above would help to highlight the importance of the proposed work.
  • In the last paragraph on page 2 it is mentioned that accident detection is treated as a classification problem in which accident classes or the event "no accident" are distinguished, which assumes a multiclass classification. However, in the first paragraph on page 3 it is mentioned that in the last step of the framework the distinction between "accident" and "non-accident" events is made, which implies a binary classification. The wording should be modified to clarify from this point if the proposed work handles multiclass or binary classification.
  • On page 3, in the list of contributions, in the first point it is mentioned that the work applies to cars regardless of the model and year of manufacture. This is a very open statement since the definition of "car" involves different types of vehicles. In addition to indicating that the year of manufacture does not matter, then is it possible to apply this work to cars older than a century, for example? Restructure this part of the wording to remove those ambiguities. It could be indicated that cars are required to meet certain minimum characteristics.

Related Work

  • To highlight the importance of the proposed work in comparison with the related works, it is important to mention in a general way the results obtained in the cited works, since many are mentioned but superficially.
  • An important aspect in the proposed work is the evaluation of different techniques for feature extraction, but this topic is not included in the mentioned works, with the exception of the last work cited in the last paragraph of page 6 and the first part of page 7.

Materials and Methods

  • At the beginning of page 9, the feature extraction methods used are mentioned, indicating that they are the most common methods in the state of the art, but nothing is mentioned about them in the related work section.
  • On page 10, at the end of section 3.3.1 it is mentioned that to prevent over-fitting, after HC feature extraction a feature selection process is performed using the RFE approach. Why was this method chosen? Were tests done with other feature selection methods?
  • At the beginning of section 3.4 it is mentioned that the C-SVM and RF classifiers are chosen and the reasons are indicated, but no works are cited where the characteristics mentioned for those classifiers are verified.

Results

  • Sections 4 and 5 are very short and are linked, so it is possible to put them together in a single section with the title "Experiments and results" so that they complement each other.

Analysis

  • On page 17, why is it said that the features learned by DNNs are difficult to interpret?
  • At the end of page 17 the signals that have the greatest impact on the result are mentioned: steering angle, speed and acceleration, in that order. This is an important discovery and the analysis in this part of the section should be further developed.

Conclusion

  • It would be interesting to include as future work the way in which a robust feature selection process could help in the proposal of this work, with the aim of improving classification performance.

Round 2

Reviewer 2 Report

The work has been improved significantly, tackling all major issues and most of the minor issues that was previously pointed out.

I recommend accept in present form (minor spelling revisions before publishing).